# Using Explainable Artificial Intelligence to Identify Key Characteristics of Deep Poverty for Each Household

**Wenguang Zhang [1], Ting Lei [1], Yu Gong [1], Jun Zhang [2] and Yirong Wu [3,\*]**

[1] The Rural Governance Research Center, School of Government, Beijing Normal University, Beijing 100875, China

[2] Department of Electrical Engineering & Computer Science, University of Wisconsin Milwaukee, Milwaukee, WI 53211, USA

[3] Institute of Advanced Studies in Humanities and Social Sciences, Beijing Normal University, Zhuhai 519087, China

\* Correspondence: yrwu@bnu.edu.cn; Tel.: +86-15919263005

**Abstract:** The first task for eradicating poverty is accurate poverty identification. Deep poverty identification is conducive to investing resources to help deeply poor populations achieve prosperity, one of the most challenging tasks in poverty eradication. This study constructs a deep poverty identification model utilizing explainable artificial intelligence (XAI) to identify deeply poor households based on the data of 23,307 poor households in rural areas in China. For comparison, a logistic regression-based model and an income-based model are developed as well. We found that our XAI-based model achieves a higher identification performance in terms of the area under the ROC curve than both the logistic regression-based model and the income-based model. For each rural household, the odds of being identified as deeply poor are obtained. Additionally, multidimensional household characteristics associated with deep poverty are specified and ranked for each poor household, while ordinary feature ranking methods can only provide ranking results for poor households as a whole. Taking all poor households into consideration, we found that common important characteristics that can be used to identify deeply poor households include household income, disability, village attributes, lack of funds, labor force, disease, and number of household members, which are validated by mutual information analysis. In conclusion, our XAI-based model can be used to identify deep poverty and specify key household characteristics associated with deep poverty for individual households, facilitating the development of new targeted poverty reduction strategies.

**Keywords:** explainable artificial intelligence technology; poverty identification; deep poverty; mutual information

## 1. Introduction

Poverty eradication is a major issue for developing countries around the world. In 2020, there were more than 1.3 billion people facing poverty across 107 developing countries [1]. Developing countries and many kinds of organizations have devoted great efforts to addressing this issue through social development. With the progress of poverty alleviation, it is especially necessary to strengthen the analysis of poverty alleviation for different types of poor people, especially for those in deep poverty. This is conducive to investing resources to tackle one of the most challenging tasks in poverty alleviation, and will help the deeply poor population to achieve prosperity together.

The first task of poverty eradication is the accurate identification of poor households in order to facilitate resource allocation and policy performance evaluation [2–9]. Therefore, special attention has been paid to the identification of poor households. The traditional identification approach is to use household income, either through self-reporting or third-party sources such as tax records, to identify poor households [8,10]. However, in practice it is difficult to obtain accurate information about income in developing countries due to limited

administrative capabilities in income reporting and auditing [11]. Alternatively, considering the multidimensional nature of poverty, Alkire and Foster have developed a multidimensional poverty index to identify the poor [12]. Despite these proposed approaches, there is no consensus on how to choose appropriate parameters in index calculations [10,13].

Artificial intelligence (AI) technologies provide a robust approach to poverty identification [14] in which multidimensional household characteristics, including basic demographic information and risk factors associated with poverty, are used to estimate the probability of being in poverty. Additionally, important household characteristics can be specified so that related poverty reduction strategies can be developed. Nevertheless, ordinary models developed for the probability estimation often encounter complications in to the pursuit of high estimation accuracy. It is hard to understand the internal structure of the models, which impacts their wider application. It is necessary to investigate how the probability is estimated, that is, how the probability changes with different factors for each household. Moreover, the ranking of household characteristics is often performed on the level of poor households as a whole, not that of individual households [11,13].

In this study, we focus on developing an explainable AI-based (XAI) [15–17] model to identify deeply poor households and analyze household characteristics associated with deep poverty. This model provides better identification performance than traditional AI technologies such as logistic regression, and can be used to achieve accurate identification of poor households. Additionally, important household characteristics associated with deep poverty are specified and ranked for each household, which can be used to develop poverty alleviation measures tailored to individual households. Moreover, the mechanisms used to generate these superior results can be explained to non-specialists, which can promote wide use of the model in poverty alleviation. Our contributions are as follows:

(1) A deep poverty identification model based on the latest XAI technologies is proposed. This model can provide higher identification performance and better explainability than traditional AI technologies.

(2) A method that can identify important household characteristics associated with deep poverty for each rural household is developed.

(3) Taking all poor households into consideration, common important characteristics that can be used to identify deeply poor households are specified, which include household income, disability, village attributes, lack of funds, labor force, disease, and number of household members,

(4) A recent and validated dataset obtained from the field monitoring and investigation of poor households in 25 Chinese provinces in 2019 is prepared and utilized.

The rest of the article is organized as follows. Related works are reviewed in Section 2, including multidimensional poverty, traditional poverty identification methods, deep poverty identification, and XAI technologies. In Section 3, our materials and methods are described. Three models are introduced for deeply poor identification, including our XAI-based model, a logistic regression-based model, and an income-based model. Our results are reported in Section 4. Discussion and conclusions are provided in Section 5.

## 2. Related Work

In this section, after briefly introducing the multidimensional nature of poverty, we describe several studies related to poverty identification using variables from these multiple dimensions. In order to address potential issues due to association complexity among multiple variables, we analyze several traditional poverty identification models, including the most widely used, logistic regression. We then survey several studies related to deep poverty identification. Finally, we discuss the work related to XAI technology and its potential to further improve identification performance, and specify important characteristics associated with deep poverty for each poor household.

## 2.1. Multidimensional Poverty

The nature of poverty is multidimensional, depending on the interaction of various factors at the individual, family, and community levels [18,19]. One of the pioneers in the field of multidimensional poverty study, Amartya Sen, found that poverty is characterized not merely by a lack of income, but by a lack of drinkable water, roads, sanitary facilities, and other conditions [20,21]. Based on Sen's work, Alkire and Foster developed a multidimensional poverty index to identify the poor [12]. At present, the Global Multidimensional Poverty Index (GMPI), proposed by the United Nations Development Programme (UNDP), is the most well-known poverty identification index in the world [1,22]. It complements traditional monetary poverty measures by capturing acute deprivations in health, education, and living standards. The World Bank proposed a multidimensional poverty measure in 2018 [23] which includes three dimensions: monetary living standards, education, and access to basic services. In the Targeted Poverty Alleviation (TPA) practice in China [9], the identification of poor households takes family income as the main standard and comprehensively considers housing, education, health, and other measures. Scholars have constructed multidimensional indices to identify poor households, using variables from four dimensions [2,7], five dimensions [8,13], or six dimensions [9]. In summary, due to the multidimensional nature of poverty, the identification of poor households should rely on variables from multiple dimensions. Nevertheless, when multiple variables are available for poverty identification, it is necessary to develop models to deal with complicated association issues among variables. Additionally, the most important variables should be specified in order to achieve effective poverty alleviation by providing targeted assistance measures [8,13] Moreover, it is necessary to develop an approach for explaining how poverty identification results are obtained.

## 2.2. Traditional Poverty Identification Methods

Generally, there are two kinds of poverty identification methods, qualitative and quantitative. Qualitative poverty identification methods are utilized to ask the family heads whether they consider themselves poor [24]. in TPA practice in China, the identification is generally carried out in one of three ways. First, poor households can report their difficulties to authorities to have their poverty status recognized. Second, they can seek help from grassroots officials for reporting. Third, government departments can learn about household situations through data, such as sudden changes in medical expenditures or employment status.

Quantitative poverty identification methods can provide more reliable results than the qualitative methods when data are collected from multiple sources. As a quantitative method, Alkire and Foster's method has been widely used to identify poor households when the number of household characteristics associated with poverty is limited [4,7,8]. When the number of household characteristics is large, it is challenging to specify proper values for the poverty dimensions, indicators, cutoffs, and weights required for index calculation. In order to sidestep this challenge, AI-based methods have been used to identify the poor; these include linear regression models [5], logistic regression models [5,9,11,13,18,25,26], probit models [2–4,10], principal component analysis [11], and neural networks [14]. Among these, the logistic regression model is the most widely used in poverty identification. This model is a generalized linear model, ensuring that results take values within a meaningful probability range. However, regression models, including logistic regressions, have many issues. On the one hand, when dealing with multivariate or nonlinear association problems, regression models are prone to under-fitting or over-fitting, which can lead to inaccurate results. On the other hand, if relying on coefficients to analyze poverty determinants, regression models can identify important overall characteristics for all households, but not specific characteristics for each household, which can impact the formulation of accurate poverty alleviation strategies for individual families. Therefore, it is necessary to look for poverty identification methods that are more accurate than regression

models and which have the capability of identifying important characteristics associated with poverty for each household.

### 2.3. Deep Poverty Identification

It is important to identify deeply poor families due to the fact that deeply poor families generally have more severely disabled people, seriously ill patients, elderly people with disabilities, mentally disabled people, etc. There have been studies on the identification of Dibao households in rural areas, a kind of deeply poor households in China. Most studies have found that identification performance is low when only household income is used to identify Dibao households [2–4]. In order to improve identification performance, several studies have developed multidimensional poverty measurements in which different kinds of household characteristics are utilized to identify Dibao households. Deng and Wang identified Dibao households using logistic regression [25]. They found that identification performance was affected by the family head's age, family fixed assets, family income, education level, number of family members, whether they suffered from serious diseases, and education expenditure. Han and Gao used a probit regression model to explore the characteristics of rural Dibao families. They found that certain monetary and non-monetary characteristics were statistically significant predictors of Dibao families [10]. Zhu and Li used a probit regression model to identify Dibao households [2]. They found that, in addition to family income, the determination of rural Dibao households typically refers to other poverty characteristics such as family structure, illness, and natural disasters. In short, even though there have been studies on the identification of deeply poor households, the results of important characteristics associated with deep poverty for each household are seldom reported.

### 2.4. Explainable Artificial Intelligence (XAI) Technology

AI technology based on big data and deep learning has greatly improved the accuracy of prediction models, finding uses in fields of application such as finance, medical care, transportation, manufacturing, and social good [27,28]. However, the biggest problem is that the algorithms increasingly used in prediction models are "black boxes", making it a challenge to explain their decisions and actions to human users. XAI technology has recently been developed to ensure the accuracy of prediction models and improve the transparency of the model operation by evaluating the importance of each feature variable for each sample in the prediction [29–34]. In the analysis of household characteristics for poverty identification, XAI technology can accurately identify deeply poor households and specify important characteristics for each poor household, which is beneficial in the development of household-based poverty alleviation strategies.

## 3. Materials and Methods

In this section, we first introduce the procedures used to perform the survey and collect the data. The results of data analysis are presented as well. Second, we describe the procedures used to prepare the variables used in deep poverty identification. Third, we develop an XAI-based poverty identification model. Its dependence plot, summary bar chart, and decision plot are described. For performance comparison, we develop a logistic regression model and an income-based model. Fourth, to validate the results of important household characteristics for deep poverty identification, we introduce mutual information technology. Finally, a metric related to the Receiver Operating Characteristic curve is introduced to compare identification performance.

### 3.1. Data

Under the leadership of the Poverty Alleviation and Development Leading Group (PADLG) of the State Council, the Rural Governance Research Center at the Beijing Normal University School of Government organized a team of over 500 people to carry out the on-site monitoring and investigating of the registered poor households located in three eastern

provinces and 22 central/western provinces in China. According to the requirements of the PADLG, the total number of households to be surveyed should be more than 35,000. In addition, a maximum of five villages can be surveyed in each poor county and up to five team members can perform the survey for one day in each village. Assuming that the task volume is about ten questionnaires per person per day, the team calculated that 250 households could be surveyed in each county and that they should conduct surveys in about 140 counties. Hence, during the survey, based on the total number and distribution of poor counties in each province, the team randomly selected a total of 146 poor counties. Among these, only one county was selected from each of Xinjiang and Tibet due to geographical constraints. In each poor county, the team selected five villages with different levels of poverty based on factors such as the incidence of poverty and the population that had not been lifted out of poverty. In each village, the team randomly selected about 50 households and met with the head of each in order to collect related information. In total, the data included 12,967 non-poor households and 23,335 poor households.

　　　The team designed a data collection form based on the requirements of "one standard, two no-worries, and three guarantees" [35] and the data items in the National Poverty Alleviation and Development Information System, which contains the location of the village, production and living conditions, village attributes, the name of the head of household, the number of household members, the number of labor forces, the reasons for being in poverty, poverty alleviation measures, income, transportation, housing, etc. For the 23,335 poor households, the team collected data using the data collection form during the survey and checked the accuracy of the information on the registration record with the data from the data collection form. After receiving the data from the team, we found that the data from 28 households lacked information on the reasons for being in poverty. After their removal, we used the data from the remaining 23,307 poor households to carry out the research. The distribution of the 23,307 poor households located in 25 provinces is shown in Table 1. The distributions of the number of household members and members of the labor force are shown in Figures 1 and 2, respectively.

**Table 1.** Distribution of sampled poor households in 25 provinces.

| Province | Households |
|:---:|:---:|
| Shandong | 792 |
| Liaoning | 677 |
| Fujian | 318 |
| Hebei | 711 |
| Shanxi | 1093 |
| Inner Mongolia | 459 |
| Jilin | 333 |
| Heilongjiang | 463 |
| Anhui | 1068 |
| Jiangxi | 753 |
| Henan | 959 |
| Hubei | 1526 |
| Hunan | 1349 |
| Guangxi | 2143 |
| Hainan | 474 |

**Table 1.** *Cont.*

| Province | Households |
|---|---|
| Chongqing | 326 |
| Sichuan | 1329 |
| Guizhou | 1893 |
| Yunnan | 2062 |
| Shaanxi | 986 |
| Gansu | 1337 |
| Qinghai | 455 |
| Xinjiang | 1016 |
| Ningxia | 574 |
| Tibet | 211 |

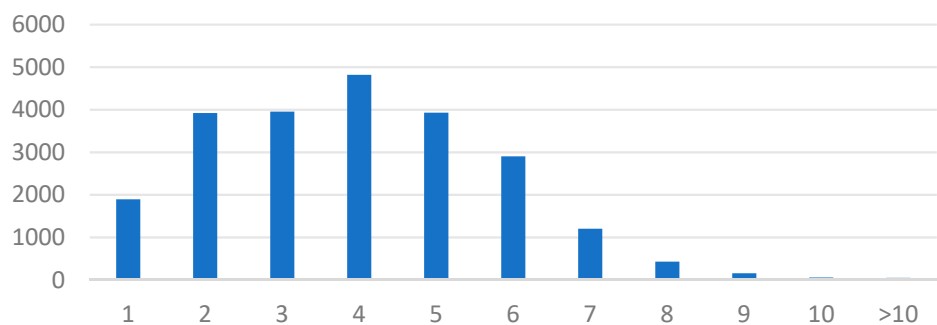

**Figure 1.** Distribution of the number of household members.

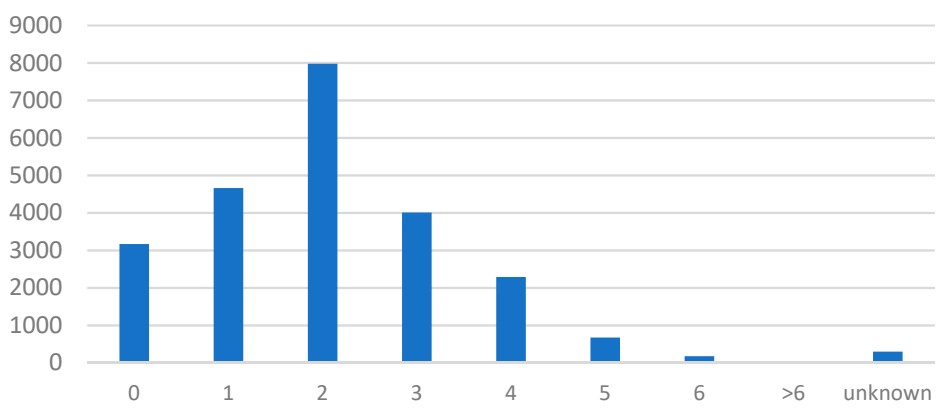

**Figure 2.** Distribution of the number of labor force participants.

*3.2. Variables*

In order to carry out deep poverty identification and analysis of important household characteristics associated with deep poverty, we selected the attributes of villages, the number of household members, the number of labor force participants, the total income of the family, their reasons of being in poverty, and the property of poor households. Input variables to deep poverty identification models are described and prepared as follows.

(1) There were four types of village attributes in this study: non-poor villages, out-of-poverty villages, poor villages, and extremely poor villages.

(2) Based on the number of household members, poor households were divided into four types of households: 1–2 persons, 3–4 persons, 5–6 persons, and households with more than 6 persons, with four-person households accounting for the largest proportion at 20.67%.

(3)　Based on the number of labor force participants in each family, poor households were divided into five types of households: 0 labor force, 1–2 labor force, 3–4 labor force, more than 4 labor force, and unknown, with two-laborer families accounting for the highest proportion at 34.24%.

(4)　According to the annual household income, poor households were divided into six types of families: less than CNY 10,000 (Chinese Yuan, 1 Chinese Yuan = 0.154 American Dollar), CNY 10,000–20,000, CNY 20,000–30,000, CNY 30,000–40,000, CNY 40,000–60,000, and more than CNY 60,000 income families.

(5)　From the reasons for being in poverty of poverty, we extracted thirteen recorded poverty factors, including death, marriage, study, disability, disaster, illness, lack of land, lack of technology, lack of water, lack of funds, inconvenient transportation, lack of self-development motivation, and other reasons.

In the end, seventeen variables associated with deep poverty were obtained. The distribution of the seventeen variables and their values is shown in Table 2. In China, poor households can be divided into extremely poor and supported households (extremely poor households), low-income households enjoying the minimum living guarantee (Dibao households), households that have been lifted out of poverty and still enjoy the poverty alleviation policy (out-of-poverty households), and generally poor households. Due to their similarity in terms of the severity of poverty and the need for a living guarantee, both extremely poor households and Dibao households can be classified as "deeply poor" households, while out-of-poverty households and generally poor households are classified as non-deeply poor households [36]. In total, there were 5442 deeply poor households and 17,865 non-deeply poor households in the dataset used in this study.

**Table 2.** Distribution of seventeen feature variables and their values.

| Feature Variables | Feature Values | Non-Deeply Poor Households | Deeply Poor Households |
|---|---|---|---|
| Number of household members | 1–2 | 3888 (66.90%) | 1924 (33.10%) |
| | 3–4 | 7046 (80.32%) | 1726 (19.68%) |
| | 5–6 | 5488 (80.32%) | 1345 (19.68%) |
| | More than 6 | 1443 (76.35%) | 447 (23.65%) |
| Number of labor force participants | 0 | 1858 (58.65%) | 1310 (41.35%) |
| | 1–2 | 9651 (76.33%) | 2992 (23.67%) |
| | 3–4 | 5390 (85.53%) | 912 (14.47%) |
| | More than 4 | 748 (83.58%) | 147 (16.42%) |
| | unknown | 218 (72.91%) | 81 (27.09%) |
| Disability | yes | 1,737 (60.15%) | 1151 (39.85%) |
| | no | 16,128 (78.99%) | 4291 (21.01%) |
| Lack of land | yes | 620 (72.94%) | 230 (27.06%) |
| | no | 17,245 (76.79%) | 5212 (23.21%) |
| Illness | yes | 5419 (71.70%) | 2139 (28.30%) |
| | no | 12,446 (79.03%) | 3303 (20.97%) |
| Lack of self-development motivation | yes | 1828 (78.42%) | 503 (21.58%) |
| | no | 16,037 (76.45%) | 4939 (23.55%) |
| Lack of technology | yes | 8459 (80.58%) | 2039 (19.42%) |
| | no | 9406 (73.43%) | 3403 (26.57%) |
| Inconvenient transportation | yes | 1736 (78.80%) | 467 (21.20%) |
| | no | 16,129 (76.43%) | 4975 (23.57%) |
| Study | yes | 2551 (80.93%) | 601 (19.07%) |
| | no | 15,314 (75.98%) | 4841 (24.02%) |

**Table 2.** *Cont.*

| Feature Variables | Feature Values | Non-Deeply Poor Households | Deeply Poor Households |
|---|---|---|---|
| Death | yes | 11 (68.75%) | 5 (31.25%) |
| | no | 17,854 (76.66%) | 5437 (23.34%) |
| Other reasons | yes | 122 (80.26%) | 30 (19.74%) |
| | no | 17,743 (76.63%) | 5412 (23.37%) |
| Disaster | yes | 451 (82.00%) | 99 (18.00%) |
| | no | 17,414 (76.52%) | 5343 (23.48%) |
| Lack of funds | yes | 6521 (84.43%) | 1203 (15.57%) |
| | no | 11,344 (72.80%) | 4239 (27.20%) |
| Lack of water | yes | 122 (84.14%) | 23 (15.86%) |
| | no | 17,743 (76.60%) | 5419 (23.40%) |
| Marriage | yes | 36 (90.00%) | 4 (10.00%) |
| | no | 17,829 (76.63%) | 5438 (23.34%) |
| Village attributes | Non-poor | 3041 (71.81%) | 1194 (28.19%) |
| | Out-of-poverty | 4072 (87.93%) | 559 (12.07%) |
| | Poor | 10,703 (74.39%) | 3685 (25.61%) |
| | Extremely poor | 49 (92.45%) | 4 (7.55%) |
| Total household income (CNY) | 1–10,000 | 1986 (59.43%) | 1356 (40.57%) |
| | 10,001–20,000 | 4762 (71.54%) | 1894 (28.46%) |
| | 20,001–30,000 | 4415 (79.68%) | 1126 (20.32%) |
| | 30,001–40,000 | 2905 (84.52%) | 532 (15.48%) |
| | 40,001–60,000 | 2514 (86.93%) | 378 (13.07%) |
| | >60,000 | 1283 (89.16%) | 156 (10.84%) |

*3.3. The XAI-Based Model*

With the rapid development of AI in recent years, machine learning models represented by deep learning and ensemble learning have played an increasingly important role in various application fields. When pursuing high-precision models with complex algorithms, industry and academia should obtain a more intuitive explanation of the results. As a classic XAI framework, SHapley Additive exPlanation (SHAP) [31] can reasonably explain the results of complex machine learning models. SHAP introduces a measure, the Shapley value, which can be used to quantify the impact of each feature variable in each sample to explain why the result can be produced. The Shapley value was first proposed by Professor Lloyd Shapley of the University of California, Los Angeles [37], where it was mainly used to solve the distribution equilibrium problem in cooperative game theory. From the perspective of game theory, each feature variable in the dataset can be regarded as a player and the predicted value of the model can be regarded as the outcome of a game. For a single sample, the Shapley value calculated for each feature variable, that is, the degree of influence of the feature variable on the model prediction, can be understood as the contribution of each player in this round of the game.

The Shapley value satisfies a number of desirable properties, making SHAP one of the most attractive explainable artificial intelligence technologies [31]. However, the Shapley value has an exponential level of computational complexity, resulting in a slow calculation speed. Recently, scholars have proposed an approximation method, TreeExplainer [32], which can be used to quickly and accurately calculate the Shapley values. Leveraging TreeExplainer, we developed an XAI-based model to identify deeply poor households. We used TreeExplainer to calculate the Shapley value of each poverty variable for each family, which was then used to guide the implementation of accurate household-based poverty alleviation strategies.

For a dataset where the number of feature variables (household characteristics associated with poverty) is $M$ and the number of samples (poor households) is $N$, TreeExplainer can generate a series of results which can be described using the following evaluation metrics.

### 3.3.1. Dependence Plot

The dependence plot describes the *distribution* of the Shapley values of all samples for a certain feature variable. In the dependence plot, the corresponding Shapley values are on the $y$ axis and the feature values of the variable are on the $x$ axis. The dependence plot can be used to illustrate the impacts of various feature values on deep poverty identification. For instance, it can be used to demonstrate that different numbers of household members have different effects on the probability of households being identified as deeply poor.

### 3.3.2. Summary Bar Chart

The summary bar chart describes the *average* of the Shapley values of all samples for each feature variable. It can be used to rank the overall importance of feature variables in deep poverty identification. For each feature variable, the average of all Shapley values of all samples can be calculated as follows:

$$I_j = \sum_{i=1}^{N} |{}_j^i| / N \tag{1}$$

where ${}_j^i$ is the Shapley value for feature variable $j$ and sample $i$, $j = 1 \ldots M$, $i = 1 \ldots N$, and $I_j$ is the average of the Shapley values for variable $j$. In this study, the larger the average value $I_j$, the more important variable $j$ is in deep poverty identification.

### 3.3.3. Decision Plot

For a particular sample, the decision plot describes the influence of each feature variable on deep poverty identification based on the Shapley values. It explains the results of the model, and can be used to assist in decision-making. In the decision plot, the probability of being identified as deeply poor is on the $x$ axis and the feature variables are on the $y$ axis. The zigzag line from bottom to top indicates that the probability varies with different feature variables. The decision plot can be used to specify the most important household characteristics associated with deep poverty. The probability of a household being identified as deeply poor can be obtained as well.

### 3.4. *The Logistic Regression-Based Model and the Income-Based Model*

We developed a logistic regression-based model to identify deeply poor households as well. The model is described as follows:

$$\log\left(\frac{p}{1-p}\right) = \alpha + \sum_{i=1}^{M} \beta_i x_i \tag{2}$$

where $p$ is the probability of a household being identified as deeply poor, $\alpha$ is the intercept of the model, and $\beta$ is the weighting of variables (household characteristics). The odds ratio (OR) is obtained by $e^{\beta}{}_i$, which can be used to analyze the importance of household characteristics in deep poverty identification, while $M$ is the number of household characteristics. In this study, OR is used to validate the results presented in the dependence plots of our XAI based model.

The income-based model for deep poverty identification, a special case of the logistic regression model, is described as follows:

$$\log\left(\frac{p}{1-p}\right) = \alpha + \beta x \tag{3}$$

where $x$ is the total household income variable.

*3.5. Mutual Information*

Mutual information is an effective information measure in information theory [35,38]. It can be regarded as the amount of information contained in a random variable about another random variable or the reduction in uncertainty of a random variable due to the knowledge of another random variable. The less the uncertainty, the more accurate the judgment or prediction. In this study, mutual information can be expressed as the amount of uncertainty in deep poverty identification that is reduced due to knowledge of a household variable. Therefore, mutual information can be used to measure the importance of household characteristics to the identification of deep poverty. The greater the value of mutual information, the more important the household characteristics are for deep poverty identification. In this study, mutual information was used to validate the results presented in the summary bar chart of our XAI-based model.

*3.6. Receiver Operating Characteristic Curve (ROC Curve)*

In the medical domain, ROC curves have been widely used to describe the performance of a medical test. True positive rate (TPR) is defined as the proportion of people who are positive in this medical test relative to the sick population. True negative rate (TNR) is defined as the proportion of people who are negative in this medical test relative to the health population. The ROC curve is constructed with a series of TPR and TNR values by setting different thresholds for the estimated probability of disease. The curve is drawn with TPR as the ordinate and (1–TNR) as the abscissa. The larger the area under the ROC curve (AUC), the higher the accuracy of the test. The value of AUC is between 0 and 1, where 1 means the test is completely correct and 0 means the test is completely wrong. In this study, the ROC curve was constructed using the probability that households would be identified as deeply poor. The AUC was then used to quantify the accuracy of deep poverty identification models.

**4. Results**

In this section, we first report our XAI-based model's accuracy in deep poverty identification. Then, we obtain common important household characteristics associated with deep poverty for all households and compare the results between our XAI-based model and mutual information analysis. Moreover, we demonstrate that different values have unique contributions to deep poverty identification for each key characteristic. Finally, by leveraging our model's explainability we describe how household characteristics impact deep poverty identification for individual households using two examples. Accordingly, important household characteristics are specified for each household.

*4.1. Identification Accuracy of Our XAI-Based Model*

Our XAI-based model is developed for deep poverty identification and uses seventeen feature variables. The ROC curve of the model is shown in Figure 3. With an AUC of 0.733, this model can be used to identify households in deep poverty [25,39]. The logistic regression-based model and income-based model achieve identification performances of 0.719 and 0.635 in terms of AUC, respectively.

*4.2. Common Important Household Characteristics Associated with Deep Poverty*

As can be seen from the summary bar plot of our XAI-based model, total household income is the most important factor in judging whether a household may be in deep poverty (Figure 4). Disability, village attributes, lack of funds, the number of labor force participants, the number of household members, and illness all play an important role in determining whether a household is identified as deeply poor. Household characteristics such as lack of self-development motivation, inconvenient transportation, lack of land, lack of technology, low education level, and disaster have limited influence on this judgment. These results mostly corroborate the rankings of household characteristics in the mutual information analysis (Table 3).

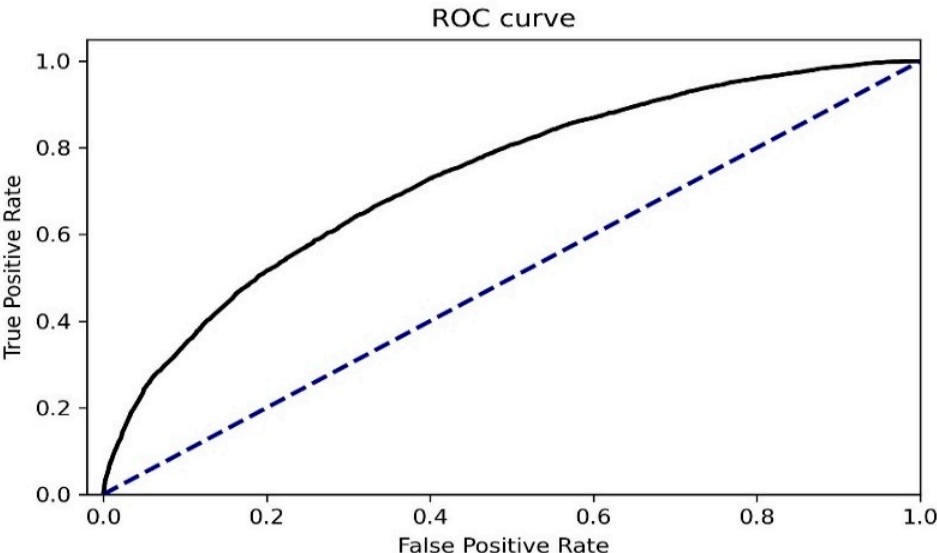

**Figure 3.** ROC curve.

*4.3. The Impact of Different Values of Each Feature Variable on Deep Poverty Identification*

In Section 4.2, we identify seven common important household characteristics associated with deep poverty, including total household income, disability, village attributes, lack of funds, the number of labor force participants, the number of household members, and illness. In this section, we analyze the impact of different feature values for each of these seven feature variables on deep poverty identification.

**Table 3.** Mutual information values and rankings of household characteristics in deep poverty identification.

| Mutual Information | Rankings | Household Characteristics |
|:---:|:---:|:---:|
| 0.0335043 | 1 | Total household income |
| 0.0260108 | 2 | number of labor forces |
| 0.0149508 | 3 | village attributes |
| 0.0139923 | 4 | disability |
| 0.0127047 | 5 | lack of funds |
| 0.0125976 | 6 | number of household members |
| 0.0051454 | 7 | lack of technology |
| 0.0046409 | 8 | illness |
| 0.0012021 | 9 | study |
| 0.0002953 | 10 | disaster |
| 0.0002031 | 11 | lack of land |
| 0.0001986 | 12 | inconvenient transportation |
| 0.0001542 | 13 | lack of water |
| 0.000148 | 14 | marriage |
| 0.0001427 | 15 | lack of self-development motivation |
| 0.0000359 | 16 | other reasons |
| 0.0000161 | 17 | death |

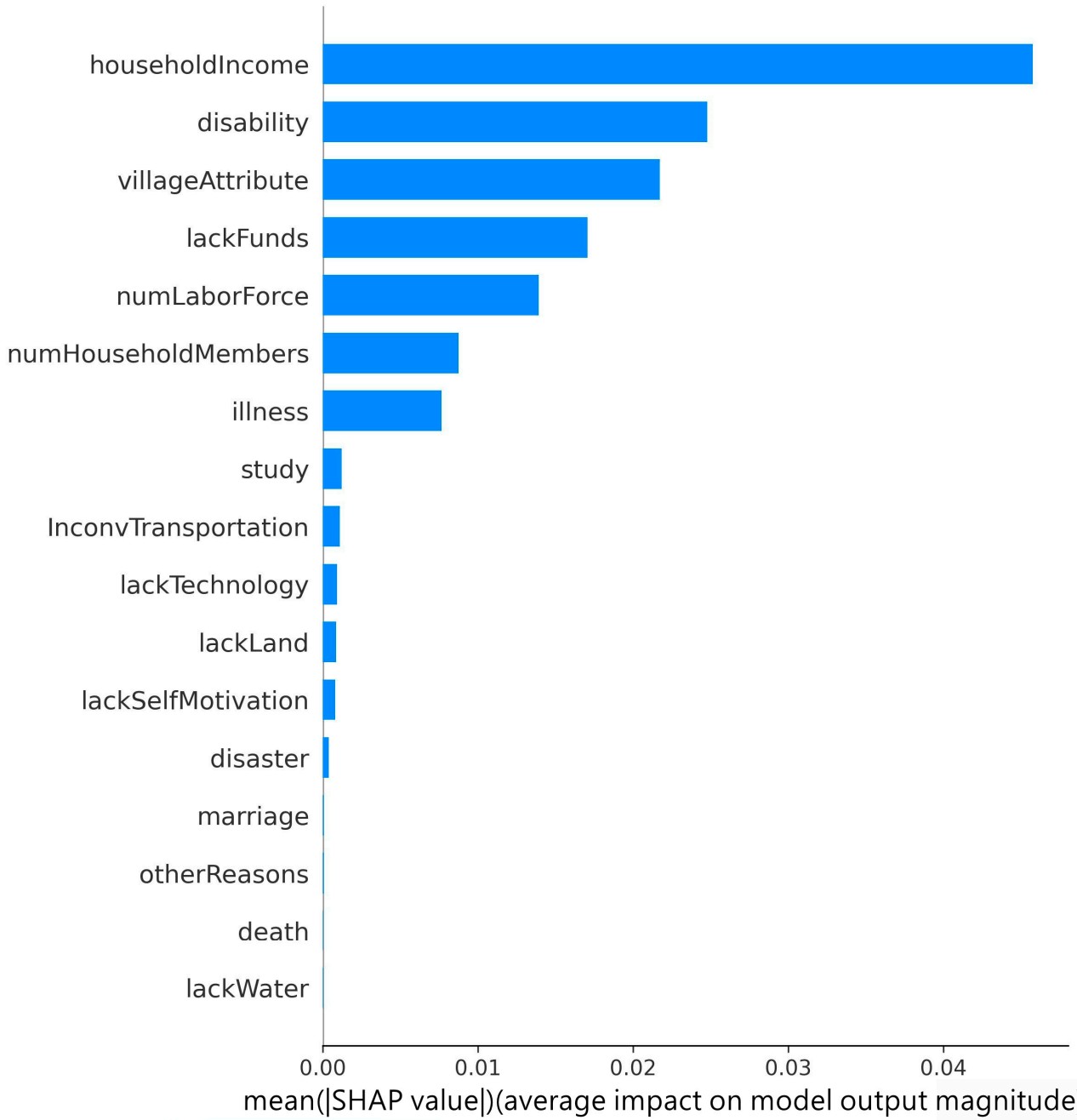

**Figure 4.** Common important household variables associated with deep poverty for all households.

4.3.1. Total Household Income

In the dependence plot (Figure 5a), we observe that as the total household income increases, the probability of the household being identified as deeply poor decreases. For poor households with an annual income of less than CNY 20,000, the Shapley values are greater than 0, indicating that these households have a higher likelihood of being categorized as in deep poverty. For poor households with an annual income greater than CNY 30,000, the Shapley values are less than 0, indicating that households with high incomes have a lower likelihood of being categorized as in deep poverty. For poor households with an annual income of CNY 20,000 to 30,000, some have the Shapley values greater than 0 and others have Shapley values less than 0, indicating that the impact of income on deep poverty identification varies from household to household. While some of these households are able to avoid deep poverty, as they may have received

certain assistance due to certain reasons, other poor households have a high chance of being identified as deeply poor. Therefore, for households with an annual income in this range, we need to further analyze the impact on households by considering other factors comprehensively.

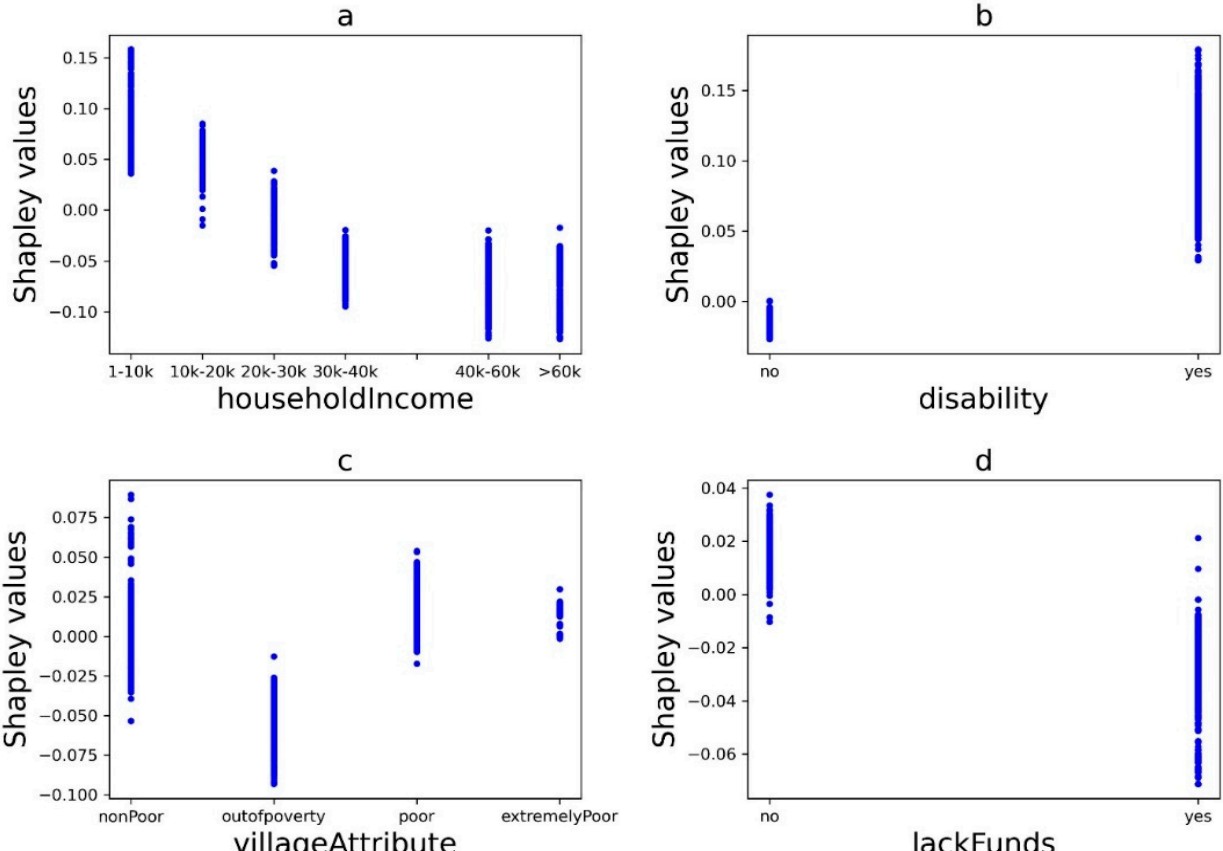

**Figure 5.** The impact of each feature variable on deep poverty identification: (**a**) total household income; (**b**) disability; (**c**) village attributes; (**d**) lack of funds.

In our odds ratio analysis, we observe that with the total household income range of CNY 1–10,000 as the base, the odds ratio is 0.682 for total household income in the range of CNY 10,000–20,000, 0.456 for total household income in the range of CNY 20,000–30,000, 0.305 for total household income in the range of CNY 30,000–40,000, 0.240 for total household income in the range of CNY 40,000–60,000, and 0.183 for total household income more than CNY 60,000. These results show that the higher the total household income, the lower the chance of being identified as deeply poor, which is in line with the results of our XAI-based model.

### 4.3.2. Disability

The Shapley values of households with disabled members are greater than 0 (Figure 5b), while the Shapley values of households without disabled members are less than 0, indicating that the chance of being identified as deeply poor for a household with disabled members is higher than for other households. A possible explanation of this increased identification chance is the combination of a disabled person's inability to work, their need for care from other family members, and the cost of treatment.

The odds ratio is 2.580 for a household with disabled members, which indicates that this kind of household has a high chance of being identified as deeply poor. This result is homologous with the results of our analysis of the Shapley values.

### 4.3.3. Village Attributes

In out-of-poverty villages, the overall poverty situation has improved thanks to the implementation of poverty alleviation measures, and the chances of households being categorized as in deep poverty are low, which is echoed by low Shapley values (Figure 5c). In non-poor villages, poor villages, and extremely poor villages, the chances of families being recognized as deeply poor are high.

Using non-poor villages as the base, the odds ratio is 0.413 for out-of-poverty villages, 1.195 for poor villages, and 0.200 for extremely poor villages. Based on analysis of these odds ratios, we observe that households in out-of-poverty villages have a low chance of being identified as deeply poor, which echoes their low Shapley values. Households in both non-poor and poor villages have a high chance of being identified as deeply poor, which corroborates the results of our analysis of the Shapley values. Households in extremely poor villages have a low chance of being identified as deeply poor, which contradicts the impact of high Shapley values. This result can be attributed to the limited number of households in extremely poor villages, indicating a need for further studies [40].

### 4.3.4. Lack of Funds

Overall, households that are poor due to a lack of funds are not likely to be identified as deeply poor households (Figure 5d). Lack of funds is not a root cause of poverty, but rather a result caused by other poverty-causing factors. In recent years, due to the development of the country's overall economy and the strengthening of poverty alleviation measures, poor households can obtain microfinance support, meaning that these households are less likely to fall into deep poverty. However, for a very few families, the Shapley values are greater than 0, indicating that lack of funds can lead to an increased chance of a household being categorized as in deep poverty.

The odds ratio is 0.601 for a household with a lack of funds, which indicates that this kind of household has a low chance of being identified as deeply poor, which is in line with the results of our analysis of the Shapley values.

### 4.3.5. Number of Labor Force Participants

As the number of labor force participants increases, the chance of being identified as deeply poor gradually decreases (Figure 6a). For households without any members in the labor force, the Shapley value is greater than 0, indicating that these households are prone to deep poverty. For households with one or two members in the labor force, certain households have an increased chance of being categorized as in deep poverty, while others have a reduced chance. Therefore, we need to analyze the impact of this variable impact on deep poverty identification by considering other factors comprehensively.

Using the value of zero labor force participants as the base, the odds ratio is 0.597 for households with one or two members in the labor force, 0.378 for households with three or four such members, 0.414 for households with more than four members, and 0.686 for households with an unknown number members participating in the labor force. These results are homologous with the results of our analysis of the Shapley values.

### 4.3.6. Number of Household Members

As the number of household members increases, the chance of being categorized as in deep poverty gradually increases (Figure 6b). For most households with three to four members, the chance of being identified as deeply poor is low. For most households with five to six members, the chance of being categorized as in deep poverty is high.

Using the value of one or two family members as the base, the odds ratio is 1.131 for households with three or 4\four family members, 1.573 for households with five or six members, and 2.650 for households with more than six members. As the number of household members increases, the chance of being categorized as in deep poverty gradually increases, which is in line with the results of our analysis of the Shapley values.

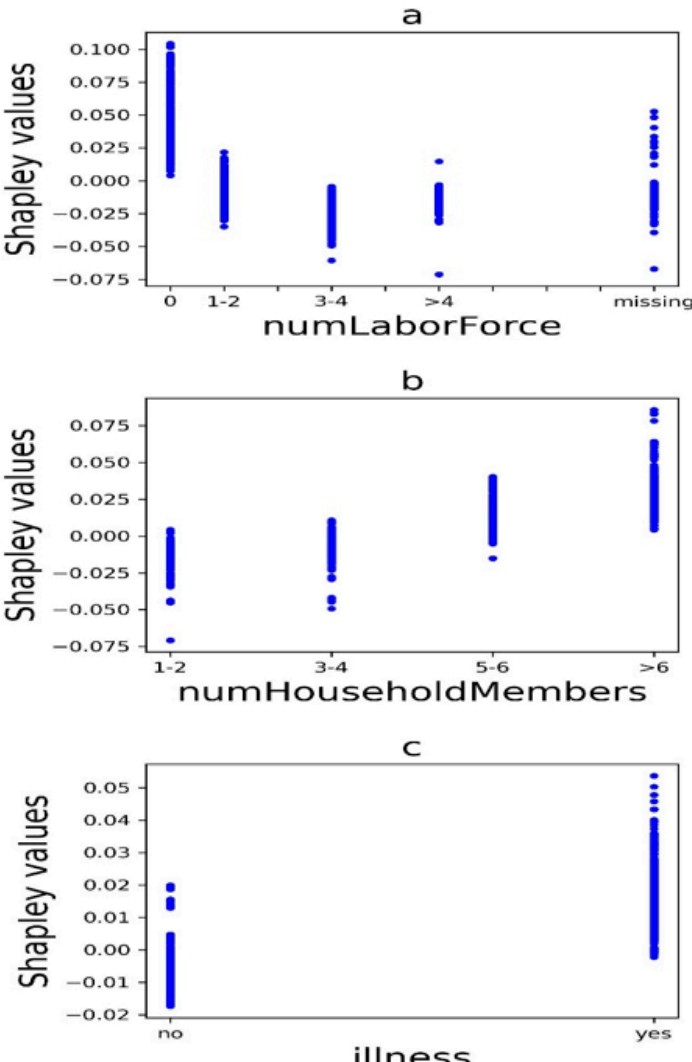

**Figure 6.** The impact of each feature variable on deep poverty identification: (**a**) number of labor force participants; (**b**) illness; (**c**) number of household members.

4.3.7. Illness

Family members' illness increases the chance of households being categorized as in deep poverty, as evidenced by the high Shapley values (Figure 6c). The odds ratio is 1.292 for a household with ill members, which indicates that this kind of household has a high chance of being identified as deeply poor.

*4.4. Key Characteristics Associated with Deep Poverty for Each Household*

**Example 1.** *An analysis of household characteristics of a non-deeply poor household.*

Based on the decision plot of our XAI-based model, we focus solely on the important characteristics at the top of the plot for a household (Figure 7). This household has five or six people (numHouseholdMembers = 5), which leads to an increased risk of deep poverty. However, this household's poverty is not related to illness (illness = 0) and the family income is between CNY 20,000 and 30,000 (householdIncome = 3), which lead to a reduction in the risk of deep poverty. The household is located in a poor village (villageAttribute = 2). Thus, the probability of the household being identified as deeply poor increases. Through further analysis, we find that the poverty of this household is not due to disability (disability = 0). A lack of funds is one of the reasons for being in

poverty (lackFunds = 1), which reduces the probability of the household being identified as deeply poor. The combined effect of these characteristics leads to the final probability of the household being identified as deeply poor falling at about 57.2%. The low probability indicates that the identification results are reliable for households that are not deeply poor.

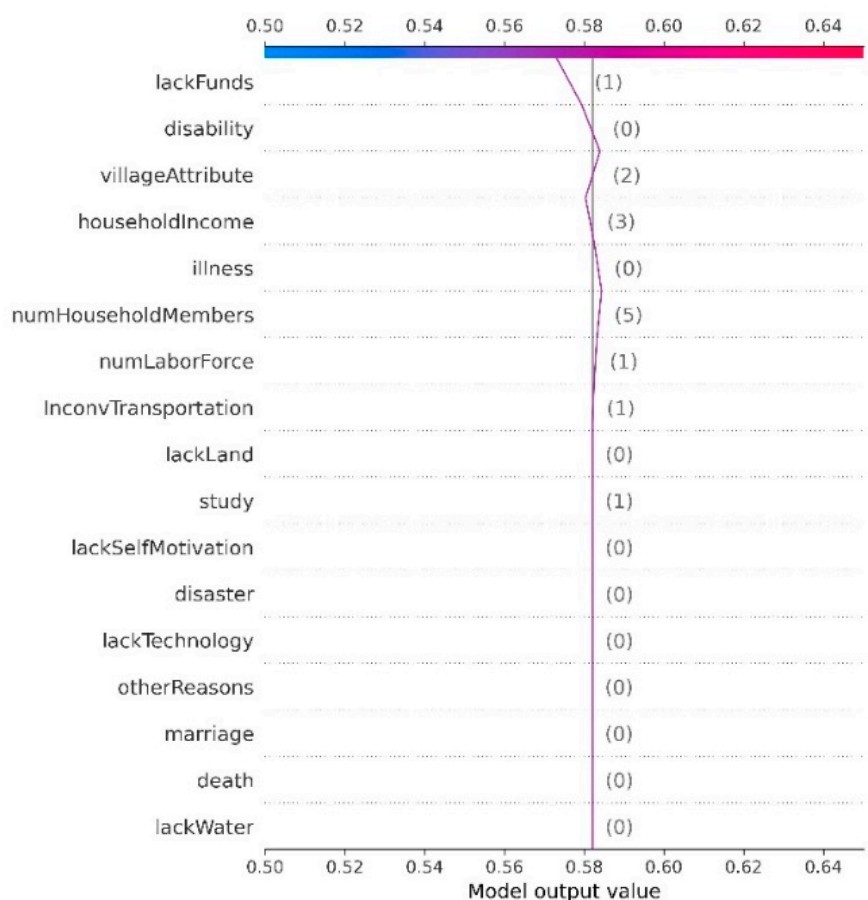

**Figure 7.** Identifying key characteristics associated with deep poverty for a non-deeply poor household.

**Example 2.** *An analysis of household characteristics for a deeply poor household.*

Similarly, we focus solely on the important variables associated with deep poverty at the top of the plot for this household (Figure 8). This household has only one to two members (numHouseholdMembers = 1), which leads to a reduction in the risk of deep poverty. The poverty of this household is not due to a lack of funds (lackFunds = 0), resulting in an increased risk of deep poverty. There is a lack of labor force participation (numLaborForce = 1), which increases the chance being identified as deeply poor. The household is located in an out-of-poverty village (villageAttribute = 1), reducing the probability of the household being identified as deeply poor. Through further analysis, we find that this household's poverty is related to disability (disability = 1) and the family income is less than CNY 10,000 (householdIncome = 1), which lead to an increased risk of falling into deep poverty. The combined effect of these characteristics leads to a final probability of the household being identified as deeply poor of about 63.0%. This high probability indicates that the identification results are reliable for deeply poor households.

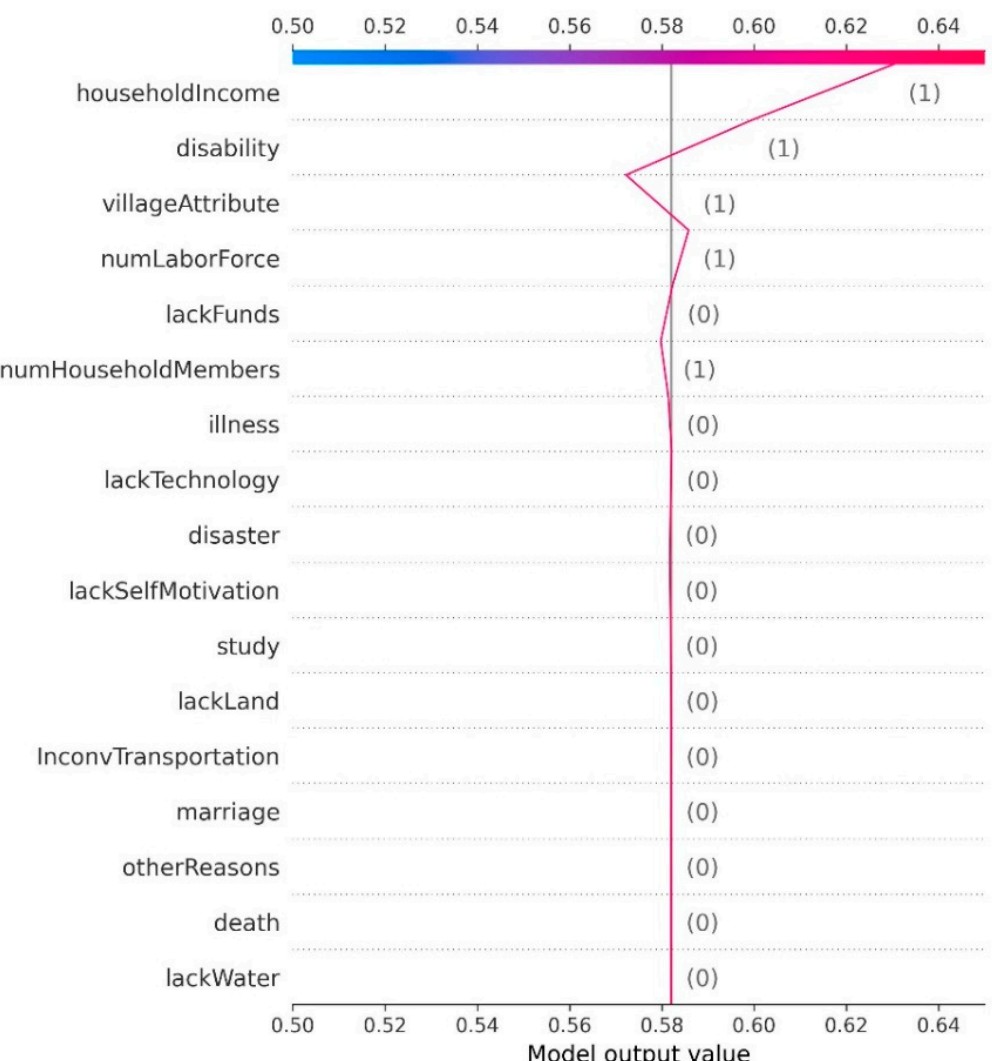

**Figure 8.** Identifying key characteristics associated with deep poverty for a deeply poor household.

### 5. Discussion and Conclusions

To improve deep poverty identification, we developed an XAI-based model to estimate the probability of households being recognized as deeply poor. We found that the XAI-based model achieves higher identification performance in terms of AUC than the logistic regression-based model and the income-based model, indicating the feasibility of establishing a model to identify poor households accurately for targeted poverty alleviation in practice. Additionally, we observed that, specific to each household, the chance of a household being identified as deeply poor changes with different key characteristics, which is conducive to the development of household-based poverty alleviation strategies for early intervention and prevention.

To the best of our knowledge, this is the first study that utilizes XAI-related methods to identify poor households. In recent years, with the gradual development and application of the Internet of Things and big data, multivariate data with complex relationships have been collected, bringing challenges to traditional AI models. To this end, experts have developed new artificial intelligence technologies such as ensemble learning and deep learning to solve such problems. However, these technologies are usually very complicated and it can be difficult to explain their internal reasons for producing accurate results. Due to this downside, XAI technology and its advantages have received increasing attention, as it can help solve complex problems involving multiple variables and nonlinearities. In the field of poverty alleviation, XAI technology can be used to identify deeply poor

households with a large number of variables. Our study involved seventeen household characteristics. The results demonstrate the feasibility of using this technology to identify key characteristics of deep poverty, laying a solid foundation for us to collect more variables to identify deeply poor households. Additionally, XAI technology can explain the results of the analysis and make the decision-making process transparent. In the field of poverty alleviation, this helps to increase the accuracy of poverty identification and to decrease the subjective impact of poverty alleviation practices, reducing the occurrence of inequities and improving transparency [41,42]. This makes XAI conducive to increasing the participation of the government and poor households in poverty alleviation, thereby achieving the desired goal of effective poverty alleviation.

The TreeExplainer method used in this study provides a unique angle to observe results that have never been observed before in the field of poverty study. Currently, most of the work relies on using relevant model parameters to identify important variables associated with poverty. The most widely used parameter is the odds ratio of the logistic regression model [18,19,25,26]. However, it cannot be used to carry out accurate poverty analysis at the individual household level. Compared with the logistic regression-based model, TreeExplainer has more advantages and helps to further promote targeted poverty alleviation. For a variable, the dependence plot can be used both to analyze the relative impact of its different values on the probability of being categorized into deep poverty and to provide the distribution of the Shapley values for all households, which presents a complete picture of the impact on poverty identification. For example, in the process of using the dependence plot to analyze the effect of total household income, we found that for households with an annual income of CNY 20,000 to 30,000, the impact of household income on deep poverty identification varies from family to family, and is not a simple number like the odds ratio. Additionally, for a poor household, the decision plot can intuitively show the process of estimating the probability of being categorized as in deep poverty based on household characteristics, helping to identify relatively important variables. The same process can be used to model the subjective decision procedure in poverty identification, helping to promote targeted poverty alleviation measures.

It is important to identify deeply poor families. For the identification of Dibao households in China, most studies have found that identification performance is low when only household income is used [2–4,10]. In this study, we found that our XAI-based model achieves higher identification performance than the income-based model, 0.733 vs. 0.635, in terms of AUC. Additionally, we found that the logistic regression model achieves higher identification performance than the income-based model, 0.719 vs. 0.635, in terms of AUC. These findings are in line with the results of previous studies [2,10]. Moreover, our XAI-based model achieves higher identification performance than the logistic regression model, 0.733 vs. 0.719, in terms of AUC, which demonstrates that it is possible to improve identification performance for Dibao households further using advanced AI technologies. Furthermore, our study found that common important characteristics that can be used to identify deeply poor households include household income, disability, village attributes, lack of funds, labor force, disease, and number of household members, which is similar to results from previous studies on Dibao household identification [2,10,25]. Finally, by leveraging the latest XAI technology, our model provides more explainable results than other deep poverty identification methods, allowing important household characteristics associated with deep poverty to be specified for individual households.

The findings in this study have important implications for future policies that address rural poverty in China and other developing countries. First, it is important to ensure that the decision-making process in the field of poverty alleviation is objective and transparent. In this study, we developed an XAI-based model for poverty identification. We found that TreeExplainer can be used both to identify the most important household characteristics for poverty identification and to demonstrate how the probability of being in poverty changes with different characteristics. By leveraging these findings, decision-makers can explain what the main variables used to make decisions are and how decisions are made

for each household in the process of poverty identification. With greater transparency, the process can then be audited to reduce inequality, furthering the desired goal of effective poverty alleviation.

Second, it is important to promote the development of models and platforms for poverty identification. While AI technologies have been introduced into poverty alleviation research, most of the work tends to rely on the coefficients of the model to study the importance of poverty causing factors, rarely discussing the models' prediction performance in poverty identification. In this study, our developed XAI-based model for poverty identification achieved superior prediction performance, indicating the feasibility of establishing a model to identify poor households in practice. Additionally, with the rapid development of big data and AI, it is possible to collect more data from civil affairs, public security, housing, education, and other departments. Based on big data, models with high performance in poverty identification can be set up and an interface-friendly platform can be developed to achieve more convenient decision-making for poverty alleviation.

Third, greater targetedness and precision of assistance measures can be achieved through poverty identification with AI technologies. In this study, we found that the XAI-based model achieves higher identification performance in terms of AUC than the logistic regression-based model and the income-based model. It is possible to improve poverty identification performance further using more advanced technologies. After poor households are identified, related assistance measures can be provided. Additionally, after the most important characteristics are recognized, corresponding assistance measures can be provided for each household to achieve efficacious poverty alleviation [35]. For example, if disability is one of the important characteristics associated with deep poverty, the corresponding assistance measure, comprehensive guaranteed poverty alleviation, should be offered to achieve poverty alleviation.

Fourth, hierarchical management should be performed for different kinds of poverty. With continued progress in poverty alleviation, it is necessary both to achieve accurate identification of poor households, and to differentiate between different root causes of poverty. This study focuses on the identification of deeply poor households, which is conducive to investing resources into tackling one of the most challenging tasks in poverty alleviation and can help the deeply poor population to achieve prosperity together. Compared with families that are not deeply poor, deeply poor households have heavier poverty-alleviation tasks and a higher risk of returning to poverty.

In this study, there are several limitations that can be addressed by future research. First, the results of this study are based on analysis of the data obtained from field monitoring and investigation of poor households in 25 provinces of China in 2019. This limitation should be considered when carrying out poverty alleviation practices in other countries. For future work, it is necessary to collect similar datasets from other countries in order to validate our results. Second, the data used in this study includes two parts: objective data, including income, family structure data, and village attributes; and subjective reasons for being in poverty, obtained via survey. Based on this data, we established our XAI-based model. The next step in related study should introduce more variables, such as household fixed assets, operating income, education expenditure, etc., to further improve the model's identification performance and provide support for targeted poverty alleviation by leveraging the model's explainability. Third, this study quantified identification performance using AUC. It is necessary to specify an optimal threshold so that the number of deeply poor households which are correctly identified can be reported. Fourth, in this study, our research into deep poverty was carried out nationwide. The next step is to conduct separate research for different regions/provinces, which is conducive to the development of targeted poverty alleviation measures [43,44].

In conclusion, our XAI-based model can achieve higher identification performance than either the logistic regression-based model or the income-based model. Additionally, for each household, key characteristics of deep poverty can be specified using our XAI-based model. The change in the probability of being identified as deeply poor with

different characteristics can be observed, allowing for the formulation of poverty alleviation strategies tailored to individual households. Important implications for future policies that address rural poverty in China and other developing countries include objective and transparent decision-making for poverty identification, development of models and platforms for poverty alleviation, pursuit of targeted poverty alleviation measures, and hierarchical management for different types of poor households [45,46].

**Author Contributions:** Conceptualization, W.Z. and Y.W.; methodology, J.Z.; formal analysis, T.L. and Y.G.; data curation, W.Z. and Y.G.; writing—original draft preparation, Y.G. and T.L.; writing— review and editing, W.Z., J.Z. and Y.W.; supervision, W.Z. and Y.W. All authors have read and agreed to the published version of the manuscript.

**Funding:** This study was supported by the "Assessment of Urban-Rural Integration and Development Mechanisms in County Areas" project (government purchase service code: B070201) of the Department of Policy and Reform, Ministry of Agriculture and Rural Affairs, China, and the "Improving the Function of Governance Systems at the County and Rural Levels" project of the National Rural Revitalization Bureau, China.

**Institutional Review Board Statement:** This study was approved by the Academic Committee of the School of Government, Beijing Normal University.

**Informed Consent Statement:** Informed consent was obtained from all subjects involved in the study.

**Data Availability Statement:** All data that support the findings of this study are available from the corresponding author upon reasonable request.

**Acknowledgments:** The authors thank the editors and anonymous reviewers for their helpful comments and suggestions.

**Conflicts of Interest:** The authors declare no conflict of interest.

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
