# Peer review of "Using Explainable Artificial Intelligence to Identify Key Characteristics of Deep Poverty for Each Household"

_sustainability, doi:10.3390/su14169872_

Round 1

Reviewer 1 Report

The subject of the article is interesting and fits in with the important issue of measuring poverty. The aim of the study was to develop an explainable AI (XAI) based model to identify deeply poor households and analyze household characteristics associated with deep poverty. Overall, I noticed that the author put a lot of effort into achieving their research goal. I appreciate their process of developing this research article.

1. The summary is correct, it contains all the most important elements.

2. Problem formulation: Problem formulation and research gap are well presented.

3. The selection, timeliness and review of the content of the literature is sufficient and adequate to the discussed issues.

4. The material and research methods have been properly selected and presented sufficiently accurately and clearly.

5. Data compiled in tables or presented in other graphic forms constitute the correct documentation of the content of the study.

6. The discussion of the results was presented correctly.

7. The conclusions were formulated correctly and are confirmed by the results and content of the study.

8. Quality of communication: This article has a certain level of readability and is fairly well written.

Author Response

Many thanks for the reviewer's comment,pls check the revised version.

Reviewer 2 Report

Dear author:
I have conducted a thorough review of your paper and have highlighted the following strengths:

•The topic is interesting, and it is also UpToDate.
However, several key areas need more work before publication. I have summarized the required changes in the hope that the feedback will be useful to you as you update the paper.

Abstract
•The research contributions should be articulated more clearly. The abstract does not seem to convey the rigor of research properly.

Introduction
-The introduction section would read better if the following were better stated and explained: focus of research conceptual/theoretical framework including relevant literature review studies that were used to conduct this research study
- The introduction section would read better if the factors of the sense-making apparatus because without highlighting them, it makes it difficult to draw a general conclusion
-The introduction section could explain the benefits of the study to the research. How and why the research advances knowledge in the field.

Literature Review
• Some sources are out of date. More recent studies should be included. Also, it should lead up to the research questions logically.
•The literature review is deficient in its addressing of the research gap and research model. It needs to be rewritten with a focus on the quality of the content.
•The theoretical background presentation could be improved by incorporating clarity and additional evidence regarding recent studies. The vague nature should be eliminated. It could be extended and expanded by including other theoretical and empirical literature which supports the importance of the study.

Methodology
•The methodology section needs more details. For example, details about demographic information, reliability and validity information, and any statistical or data analysis should be presented.
•Details about the methods of data collection, interviews, surveys, questionnaires, observation, and archival data must be included and explained in detail.
•This section should explain how the results were triangulated and discuss validity in detail
•The research steps need to be explained.

Discussion and Conclusion
•Improve the discussion section to ascertain better what is unique/novel about your findings
•Explain how the article contributes to new knowledge in the domain.
• Recommendations for practitioners are not articulated clearly.
•Update the conclusion to include the newly formulated theoretical contributions
•Summarize the key results in a compact form and re-emphasize their significance.
•Summarize how the article contributes to new knowledge in the domain.
•Provide suggestions for future studies.
•This conclusion could be worded in a manner as to emphatically motivate the academic community to get down to actionable, practical engaged scholarship.

Author Response

Response to Reviewer 2 Comments

     · Point 1:The topic is interesting, and it is also UpToDate.
However, several key areas need more work before publication. I have summarized the required changes in the hope that the feedback will be useful to you as you update the paper.
       Response1: We thank the reviewer for these encouraging remarks and help.

Abstract
    •
Point 2:The research contributions should be articulated more clearly. The abstract does not seem to convey the rigor of research properly.
     Response2: We thank the reviewer for these constructive comments. We have modified the abstract as follows,

  • “For comparison, a logistic regression-based model and an income-based model are also developed.” is added to describe the methods clearly.
  • “in terms of the area under ROC curve” is added to make the comparison of identification performance more specific.
  • “while ordinary feature ranking methods can only provide ranking results for poor households as a whole” is added to make the description of our contributions more clearly.
    With these modifications, the rigor of research would be improved.

Introduction
     •Point 3:The introduction section would read better if the following were better stated and explained: focus of research conceptual/theoretical framework including relevant literature review studies that were used to conduct this research study

     Response3: The focus of research framework is to develop an explainable AI based model to identify deeply poor households and analyze household characteristics associated with deep poverty. After the fourth paragraph of the introduction, more sentences are added to highlight the main contributions, including the explainable AI based model and the method that can identify important household characteristics associated with deep poverty for each rural household. Relevant literature is also added.

  • Point 4:The introduction section would read better if the factors of the sense-making apparatus because without highlighting them, it makes it difficult to draw a general conclusion

           Response 4: We thank the reviewer’s help. We are trying to understand what the reviewer is trying to say but because some words or phrases seem to be missing between the word "apparatus" and "because," it is difficult to tell what the reviewer is trying to say.

  • Point 5:The introduction section could explain the benefits of the study to the research. How and why the research advances knowledge in the field.

      Response 5: In the fourth paragraph of the introduction, more sentences are added to build up the connection between this study and poverty identification research. The model and method proposed in this study can be applied to wide applications in poverty alleviation practice.

Literature Review
        • Point 6:Some sources are out of date. More recent studies should be included. Also, it should lead up to the research questions logically.

        Response 6: Several recent studies are referenced and several transition sentences are added,with a hope of leading up to the research questions logically.

  • Point 7 :The literature review is deficient in its addressing of the research gap and research model. It needs to be rewritten with a focus on the quality of the content.

           Response 7: We thank the reviewer for this constructive comment. At the beginning of Section 2, a short summary of literature survey is provided to describe the connections among review topics. Additionally, several transition sentences are added to address the research gap and research model.

  • Point 8 :The theoretical background presentation could be improved by incorporating clarity and additional evidence regarding recent studies. The vague nature should be eliminated. It could be extended and expanded by including other theoretical and empirical literature which supports the importance of the study.
    Response 8: The theoretical background presentation is improved by adding a sub-section 2.3 related to deep poverty identification. Additionally, several theoretical and empirical literature is also added to support the importance of the study.

Methodology
         • Point 9 :The methodology section needs more details. For example, details about demographic information, reliability and validity information, and any statistical or data analysis should be presented.

         Response 9: The goal of our study is to identify key characteristics of deep poverty for each household. The subject of this study is poor households instead of poor people. Demographic information for households is limited, including the location, the number of household members, and the number of labor forces only. We reorganize the materials in Section 3 and present data analysis results in Table 1 Distribution of sampled poor households in 25 provinces, Figure 1 Distribution of the number of household members, and Figure 2 Distribution of the number of labor forces, respectively.

The data in this study is obtained from the event of on-site monitoring and investigating of the registered poor households located in 25 provinces in China. The purpose of this event is to check the accuracy of the information on the registration record. Reliability and validity tests have been performed internally. Additional data tests are not performed in this study.

  • Point 10:Details about the methods of data collection, interviews, surveys, questionnaires, observation, and archival data must be included and explained in detail.

        Response 10: In the Section 3.1, more materials are added to describe how the survey is performed, how the questionnaire is designed, and how the data is collected and checked.

  • Point 11:This section should explain how the results were triangulated and discuss validity in detail

         Response 11: In this study, several sentences are added to describe that the results presented in the dependence plots of our XAI based model can be validated partially by odds ratio. Additionally, mutual information can be used to validate the results presented in the summary bar chart of our XAI based model. The area under the ROC curve can be used to differentiate the performance between poverty identification models.

  • Point 12:The research steps need to be explained.

      Response12: More materials are added to explain the research steps in the first paragraph of section 3.

Discussion and Conclusion
     •
Point 13:Improve the discussion section to ascertain better what is unique/novel about your findings

      Response13: We thank the reviewer for this constructive comment. In the 3rd paragraph of Section 5, we highlight that TreeExplainer used in this study provides us a unique angle to observe some results that were never observed before in the field of poverty study. Specifically, for a variable, the dependence plot can provide the distribution of the Shapley values for all households, which presents a whole picture of the impact on poverty identification. For a poor household, the decision plot can intuitively show the process of estimating the probability of being categorized into deep poverty based on the household characteristics, helping to identify relatively important variables.

  • Point 14:Explain how the article contributes to new knowledge in the domain.

         Response14: In the 4th paragraph of Section 5, for deep poverty identification, 1) we verify that identification performance will be low when only household income is used to identify Dibao households. 2) We verify that to improve identification performance, multidimensional poverty measurements should be developed to identify Dibao households. 3) We observe that common important characteristics that can be used to identify deeply poor households include household income, disability, village attributes, lack of funds, labor force, disease, and number of household members. 4) We observe that XAI technology can be utilized to obtain more accurate and more explainable identification results than other methods.

  • Point 15:Recommendations for practitioners are not articulated clearly.

          Response15: We agree with the reviewer that recommendations for practitioners are not articulated clearly. In the revised manuscript, four recommendations have been added in the 5th, 6th, 7th, 8th paragraph of Section 5, including objective and transparent decision-making for poverty identification, development of the models and platforms for poverty alleviation, pursuit of targeted poverty alleviation measures, and hierarchical management for different types of poor households.   

  • Point 16:Update the conclusion to include the newly formulated theoretical contributions

         Response16: In the 2nd paragraph of Section 5, we highlight that to the best of our knowledge, this is the first study that utilizes XAI related methods to identify poor households. The development of the methodology of using XAI technologies in poverty alleviation is the most important contribution of our study.  

  • Point 17:Summarize the key results in a compact form and re-emphasize their significance.

        Response 17: In the 1st paragraph of Section 5, we re-prepare the results in a compact form by deleting some unimportant sentences and re-emphasize their significance.

  • Point 18:Summarize how the article contributes to new knowledge in the domain.

        Response18: The summary of the contributions is provided at the end of the 4th paragraph in Section 5.

  • Point 19:Provide suggestions for future studies.

        Response 19 : We thank the reviewer for this constructive comment. In the 9th paragraph of Section 5, after several work limitations have been described, the suggestions for future studies are provided.  

  • Point 20:This conclusion could be worded in a manner as to emphatically motivate the academic community to get down to actionable, practical engaged scholarship.

       Response20: In the last paragraph of section 5, for conclusion, after our model and its significance are described, we add several sentences to depict actionable work for poverty alleviation in practice.

Round 2

Reviewer 2 Report

I am generally satisfied with the revisions and recommend the article for publication.